# Exploring Hyperprolific Sows: A Study of Gross Morphology of Reproductive Organs and Oxytocin Receptor Distribution across Parities

**DOI:** 10.3390/ani14131846

**Published:** 2024-06-21

**Authors:** Yosua Kristian Adi, Preechaphon Taechamaeteekul, Sawang Kesdangsakonwut, Paisan Tienthai, Roy N. Kirkwood, Padet Tummaruk

**Affiliations:** 1Centre of Excellence in Swine Reproduction, Department of Obstetrics, Gynaecology and Reproduction, Faculty of Veterinary Science, Chulalongkorn University, Bangkok 10330, Thailand; yosua.kristian.a@ugm.ac.id (Y.K.A.); taechamaeteekul.p@gmail.com (P.T.); 2Department of Reproduction and Obstetrics, Faculty of Veterinary Medicine, Universitas Gadjah Mada, Yogyakarta 55281, Indonesia; 3CU-Animal Fertility Research Unit, Department of Pathology, Faculty of Veterinary Science, Chulalongkorn University, Bangkok 10330, Thailand; sawang.k@chula.ac.th; 4Department of Anatomy, Faculty of Veterinary Science, Chulalongkorn University, Bangkok 10330, Thailand; paisan.t@chula.ac.th; 5School of Animal and Veterinary Sciences, University of Adelaide, Roseworthy, SA 5371, Australia; roy.kirkwood@adelaide.edu.au

**Keywords:** aging, hormone receptor, hyperprolific sows, immunohistochemistry, uterus

## Abstract

**Simple Summary:**

Today, sows produce larger litter sizes than they did several years ago. Modern hyperprolific sows with greater parity numbers experience longer farrowing durations and require more farrowing assistance compared to those with lower parity numbers. Among reproductive hormones, oxytocin plays a critical role during parturition and is often given to sows during farrowing to shorten its duration. Oxytocin is a uterotonic hormone originating from the hypothalamus and released from the posterior pituitary gland, and it is important for the expulsion of fetuses. Aging is a well-known factor that affects productivity and increases farrowing duration in modern prolific sows. In the current study, we demonstrated a decrease in the immunolocalization of oxytocin receptors in the uterine tissue of older sows, supporting a previous study that showed weaker uterine contractions in sows with a greater parity number, which possibly affects the duration of farrowing.

**Abstract:**

This study investigated the gross morphology of reproductive organs and oxytocin receptor distribution across different parities in hyperprolific sows. A total of thirty-two reproductive organs from Landrace × Yorkshire crossbred sows were categorized into three groups based on parity numbers: 1 (*n* = 10), 2–5 (*n* = 12), and ≥6 (*n* = 10). All sows were culled due to management problems, and none had reproductive disorders. A gross morphology examination of the ovaries, uterus, and the rest of the reproductive tract was conducted. Using immunohistochemistry, the levels of oxytocin receptor were evaluated in five layers of the uterus, the epithelial, superficial glandular, deep glandular, and circular and longitudinal smooth muscles of the myometrium, and were quantified using an H-score. On average, the age and body weight of sows and the total number of piglets born per litter were 799.8 ± 327.8 days, 213.2 ± 31.7 kg, and 15.5 ± 4.8, respectively. The numbers of ovulations in sows in parity number 1 (19.9 ± 2.4) were lower than those in sows in parity numbers 2–5 (29.7 ± 2.0, *p* = 0.004) and ≥6 (27.7 ± 2.1, *p* = 0.022). The uterine weights of sows in parity number 1 (902.9 ± 112.5 g) were lower than those of parity numbers 2–5 (1442.1 ± 111.8 g, *p* = 0.001) and ≥6 (1394.3 ± 125.1 g, *p* = 0.004). The length of the uterus in sows with parity number 1 (277.9 ± 26.1 cm) was shorter than those in sows with parity numbers 2–5 (354.6 ± 25.9 cm, *p* = 0.033) and tended to be shorter than those in sows with parity numbers ≥ 6 (346.6 ± 29.0 cm, *p* = 0.068). The immunolocalization of oxytocin receptors could be detected in various parts of the porcine endometrium and myometrium. Among the five tissue layers of the uterus, the H-score of oxytocin receptors in the deep uterine glands was greater than in the superficial uterine glands (*p* = 0.023) and the circular muscle layer of the myometrium (*p* = 0.011), but it did not differ from the epithelial layer of the endometrium (*p* = 0.428) or the longitudinal muscle layer of the myometrium (*p* = 0.081). Sows with parity numbers ≥ 6 had a lower oxytocin receptor H-score than those with parity numbers 1 (*p* < 0.001) and 2–5 (*p* < 0.001). In conclusion, these data emphasize the notable variations in several reproductive parameters and the levels of oxytocin receptor within the uterus of hyperprolific sows. Across the majority of uterine tissue layers, there was a marked decrease in the H-score of the oxytocin receptor in the older sows.

## 1. Introduction

The use of prolific sows in the modern swine industry can significantly increase the number of piglets weaned per sow annually, reaching up to 30 to 40 piglets per sow per year [1]. This, in turn, enhances the overall production efficiency of a swine operation. However, in commercial swine herds, parity is a known factor that influences farrowing performance and productivity. Clinical studies have demonstrated that modern hyperprolific sows with greater parity numbers experience longer farrowing durations and require more farrowing assistance compared to those with lower parity numbers [2,3]. Additionally, it has been demonstrated that second- to fourth-parity sows have the greatest total number of piglets born per litter and the greatest number of piglets born alive per litter, considered optimal for fertility and prolificacy [4]. However, a decline in productivity is usually observed after reaching the fifth to seventh parities and decreases further in eighth to tenth parity sows [2,4]. Moreover, very old sows (≥ 8 parities) also have a greater number of stillborn piglets compared to younger ones [4]. Therefore, based on previous studies, sows can be categorized based on their reproductive potential during their lifetime into young sows with moderate productivity (gilts and primiparous), middle-aged sows with optimum productivity (parities 2–5), and old sows with decreased productivity (parity ≥ 6), while farrowing performance typically declines as parity increases [2,3,4,5,6].

It is suggested that reproductive outcomes are highly associated with several anatomical and morphological metrics of the reproductive organs in sows [7]. However, detailed information about the effect of aging, i.e., parity number, on the reproductive organs of modern hyperprolific sows is lacking, especially in those kept in tropical environments. A previous study in pigs demonstrated that the morphology of the cervix differed between multiparous and nulliparous sows, with the average dimensions of the cervix being longer and wider in multiparous sows than in nulliparous sows [8]. This reveals that the morphology of the reproductive tract may change with the parity of the female. However, information regarding the morphology of the entire reproductive tract across different parities in modern hyperprolific sows is lacking. Understanding variations in the gross morphology of reproductive organs can provide basic information that may explain differences in productivity and farrowing performance over the lifetime of sows.

Among reproductive hormones, oxytocin plays a critical role during parturition and is often given to sows during farrowing to shorten its duration. Oxytocin is a uterotonic hormone originating from the hypothalamus and released from the posterior pituitary gland. It is important for the expulsion of fetuses [9]. Within the uterus, oxytocin functions through the oxytocin receptor, a seven-transmembrane receptor on the cell surface, which triggers myometrial contractions [9]. Previous studies revealed that maternal plasma oxytocin concentrations rise at certain stages of labor, aligning with a significant surge in oxytocin receptor expression during parturition in several mammals, including rabbits [10,11], sheep [12,13], and rats [14,15]. However, measuring oxytocin during farrowing is challenging due to its pulsatile release during the phase of fetal expulsion [9]. In pigs, oxytocin receptors are expressed in both the endometrium and myometrium, not only during parturition but also throughout the reproductive cycle, providing an opportunity to evaluate them at various periods [16]. Furthermore, variations in oxytocin receptor density between the two types of myometrial muscle layers have been observed, with a greater oxytocin receptor density in the longitudinal muscle compared to that in the circular muscle types (5:1 ratio) [17]. Additionally, there was a tendency for the oxytocin receptor density to decrease from the cornua to the cervix [17]. While studies have reported the expression of the oxytocin receptor during both cyclic and gestational phases [9,17], to our knowledge, no research has been conducted to explore how aging impacts the oxytocin receptor in the pig uterus. Interestingly, however, in mice, an in vitro study of muscle stem cells revealed that oxytocin receptor expression in young satellite cells was greater than in the older ones [18]. This indicated that the level of the oxytocin receptor diminishes with the age of muscle stem cells. In contrast, a study conducted in non-pregnant mares revealed that age (2–9 years versus >10 years) did not have an impact on uterine oxytocin receptor distribution or gene expression [19]. Until now, there has been no research on the relationship between sow parity and the oxytocin receptor in various layers of their uterine tissue. In hyperprolific sows facing issues with extended farrowing durations, it is crucial to examine how the oxytocin receptor in the uterus varies with their parity number. It is possible that the oxytocin receptors in uterine tissue decrease with increasing parity number, potentially contributing to prolonged farrowing durations. Therefore, this study sought to examine the gross morphology of reproductive organs and the distribution of oxytocin receptors among sows of different parities, specifically focusing on hyperprolific sows in tropical climates, to enhance reproductive management strategies and address issues specific to each group.

## 2. Materials and Methods

### 2.1. Animals and Study Design

The Institutional Animal Care and Use Committee (IACUC) granted approval for this study, ensuring compliance with university rules and policies for the welfare and utilization of experimental animals (approval number 2331095). A total of thirty-two reproductive tracts from Landrace × Yorkshire crossbred sows were categorized into three groups based on parity numbers: 1 (*n* = 10), 2–5 (*n* = 12), and ≥6 (*n* = 10). Herd records were used to retrieve historical data of sows based on their identity numbers. All sows were culled due to management problems, none had reproductive disorders. Reproductive organs, including ovaries, oviducts, uterus, cervix, vagina, vestibule, and vulva (Figure 1), were immediately chilled on ice and transported to the laboratory within 24 h of slaughter for detailed post-mortem examination. The sows were slaughtered following standard procedures at a commercial slaughterhouse. This involved restraining them in a manner designed to minimize stress, using electrical stunning to render them unconscious and insensible to pain. Following stunning, sows were exsanguinated, after which the carcasses underwent further processing for meat production, including skinning and removal of internal organs. The reproductive performance data at the last farrowing of the sows was also collected. A post-mortem examination of the gross morphology of the ovaries, uterus, and the rest of the reproductive tract was conducted. The oxytocin receptors in the uterus of the sows were analyzed through immunohistochemistry. The levels of oxytocin receptor immunolocalization were evaluated across five layers of the uterus: the epithelial, superficial glandular, deep glandular, and circular and longitudinal smooth muscle of the myometrium, and were quantified using an H-score [20].

### 2.2. Post-Mortem Examination

Ovaries were removed from the mesovarium and then individually weighed using a digital scale (BJ 210C, Precisa Instruments Ltd., Dietikon Switzerland) with a capacity of 210 g and increments of 0.01 g. The reproductive organs were classified as in the luteal phase if the ovaries exhibited corpora hemorrhagica or corpora lutea. Conversely, they were classified as in the follicular phase if the ovaries contained follicles with a diameter of 7–12 mm, with or without corpora albicantia [21]. The number of follicles and corpora lutea in each ovary were counted. The number of ovulations was determined by summing the corpora lutea in both ovaries.

The oviducts and uterine horns were separated from the mesosalpinx and mesometrium for a detailed macroscopic examination. The uterine horn, spanning from the utero-tubal junction to the uterine body, was dissected, and its length was measured with a tape. The lengths of the uterus were calculated as the sum of both the uterine horns and the uterine body length. The uterus, including the uterine horns and uterine body, were dissected from the other parts, and their weight was determined using an electronic balance (capacity: 5 kg, measuring in 1 g increments; KD300, TANITA Corporation, Bangkok, Thailand). The uterine horns were longitudinally opened, and the condition of the endometrium, including edema and color, was assessed with consideration of the phase of the cycle. No abnormalities were detected within the uterine lumen, such as signs of endometritis (marked by severe edema, congestion, a dark red color, and, typically, purulent exudates in the lumen). The oviducts were measured for length and examined for their overall appearance and color.

The cervix, vagina, and vestibule were dissected longitudinally to confirm the absence of inflammation, including severe edema, congestion, a dark red color, and the presence of purulent exudates in the lumen. Additionally, the lengths of the cervix, vagina, and vestibule were measured using a tape.

### 2.3. Immunohistochemical Staining

The tissue samples of the uterus were fixed in 4% paraformaldehyde for 24–48 h, processed with an automatic tissue processor (Tissue-Tek VIP 5 Jr., Sakura, Tokyo, Japan), and embedded in paraffin blocks (Tissue-Tek TEC, Sakura, Tokyo, Japan). The paraffin blocks were then cut into 5 µm thick sections using a microtome (Shandon, Anglia Scientific Instrument Ltd., Cambridge, UK). Paraffin tissue sections of the uterine horn were then deparaffinized by immersing them in xylene and rehydrated by immersing them in a graded alcohol series. To restore antigenicity, the tissue sections were immersed in 0.01 M sodium citrate buffer solution (pH 6.0) in a microwave at 700 Watts for 10 min. Immunoreactivity of the primary antibody was detected using the ImmPRESS^®^ Excel Amplified Polymer Kit, Peroxidase (Anti-Rabbit IgG, MP-7601, Vector Laboratories, Inc., Newark, CA, USA). The tissue sections were then incubated in BLOXALL^®^ blocking solution for 10 min to quench endogenous peroxidase activity and subsequently incubated with normal horse serum (2.5%) for 30 min. Excess serum was removed from the tissue sections, and the tissue sections were immunolocalized for the oxytocin receptor using rabbit polyclonal antibodies (1:300 dilution in PBS, bs-1314R, Bioss Antibodies, Beijing, China) applied at 4 °C overnight in a humid chamber. The primary antibody was replaced with a phosphate buffer saline (PBS) solution as a negative control. The tissue sections were incubated with the Amplifier Antibody (Goat Anti-Rabbit IgG) for 15 min and then incubated with the ImmPress^®^—HRP Horse Anti-Goat IgG Polymer Reagent for another 30 min. Subsequently, the tissue sections were incubated in 3,3′-Diaminobenzidine (ImmPACT^®^ DAB EqV) working solution for 2 min until the desired stain intensity developed. Counterstaining was performed using Meyer’s Hematoxylin, and the sections were mounted with mounting medium (Permount^®^, Fisher Scientific, Göteborg, Sweden).

Immunohistochemistry-stained tissue imaging was captured using a digital slide scanner and evaluated using QuPath-0.5.0 and ImageJ-1 software [22]. Additionally, 3,3′-Diaminobenzidine (DAB) intensity in various regions, including the surface epithelial cells, superficial uterine glands, deep uterine glands, circular layer of the myometrium, and longitudinal layer of the myometrium, was quantified using the H-score method. The H-score was calculated as follows: (1 × percentage of weak staining) + (2 × percentage of moderate staining) + (3 × percentage of strong staining) [20]. Twenty-five areas of view (each equal to 8 × 10^4^ μm^2^) were captured and evaluated for their DAB intensity according to the standard for negative, weak, moderate, and strong staining intensity. The H-score values of the regions of interest were presented, ranging from 0 to 300. Researchers achieved consensus in cases of discrepancies.

### 2.4. Statistical Analyses

All statistical analyses were performed using SAS software version 9.4 (SAS Institute Inc., Cary, NC, USA). Descriptive statistics for continuous data, including reproductive performance, litter traits, and gross morphology measurements of the reproductive organs, were calculated using the MEAN and FREQ procedures. Spearman’s correlation analysis was conducted to determine the association between sow age (day) and body weight (kg) with various reproductive organ morphologies, including ovary weight (g), oviduct length (cm), uterus weight (g), and the number of ovulations. The continuous data on reproductive performance and gross morphology measurements of the reproductive organs were analyzed using the general linear model (GLM) procedure. The factors included in the statistical model were parity numbers (1, 2–5, and ≥6) and stages of the estrous cycle, i.e., luteal phrase (*n* = 26) and follicular phase (*n* = 6). Additionally, the H-score of the oxytocin receptor was analyzed using the GLM procedure, employing parity numbers, stages of the estrous cycle, uterine tissue layers (surface epithelial cells, superficial uterine glands, deep uterine glands, the circular layer of the myometrium, and the longitudinal layer of the myometrium), and their interactions. The least-square means and standard error of the mean (SEM) were obtained for each factor class and compared using the least significant difference (LSD) test. Statistical significance was established by assessing differences and correlations at a significance level of *p* < 0.05.

## 3. Results

### 3.1. Descriptive Statistics

Descriptive statistics on sow data, reproductive performances at the last farrowing, and the gross morphology of the reproductive organs are presented in Table 1. Additionally, Spearman’s correlations between the age and body weight of sows and their reproductive performance at the last farrowing and the gross morphology of their reproductive organs are presented in Table 2.

### 3.2. Sow Data and Reproductive Organs Morphometry

Sow reproductive performance at their last farrowing and the morphometry of sow reproductive organs among parity groups are presented in Table 3 and Table 4, respectively. The body weight of sows at culling did not differ among parity groups (*p* = 0.725) (Table 3). On average, the ovary and uterus in sows with parity numbers 2–5 and ≥6 were heavier than those in sows with parity number 1 (*p* < 0.05) (Table 4). Sows with parity numbers 2–5 and ≥6 also had a greater number of ovulations than those with parity number 1 (*p* < 0.05) (Table 4). The length of the oviduct in sows with parity numbers ≥ 6 was longer than that in sows with parity numbers 1 and 2–5 (*p* < 0.05) (Table 4). The length of the uterus in sows with parity numbers 2–5 was longer than those in sows with parity number 1 (*p* = 0.033) but did not differ from those in sows with parity number ≥ 6 (*p* = 0.817) (Table 4).

The gross morphology of reproductive organs in the luteal phase and follicular phase is presented in Table 4. A significant difference was found only in the weight of the uterus, with the uterus in the follicular phase group being heavier than those in the luteal phase group (*p* < 0.001). The other parameters were not significantly different.

### 3.3. Oxytocin Receptor in the Uterine Tissue

The immunolocalization of the oxytocin receptor could be detected in various parts of the endometrium and myometrium, including the surface epithelial cells of the endometrium, endometrial glandular cells, myocytes of the myometrium, vascular endothelial cells, vascular smooth muscles, and endometrial stromal cells of uterine tissues (Figure 2). Among the five tissue layers of uterine tissue, the H-score of oxytocin receptors was greater in the deep uterine glands compared to the superficial uterine glands (*p* = 0.023) and the circular layer of the myometrium (*p* = 0.011), but did not differ from the epithelial layer of the endometrium (*p* = 0.428) or the longitudinal layer of the myometrium (*p* = 0.081). Sows with parity numbers ≥ 6 had a lower oxytocin receptor (H-score) than those with parity numbers 1 (*p* < 0.001) and 2–5 (*p* < 0.001) (Table 5). Furthermore, compared to sows with parity numbers 2–5, sows with parity numbers ≥ 6 exhibited a lower H-score of oxytocin receptor in the superficial uterine glands (*p* = 0.013), deep uterine glands (*p* = 0.007), the circular layer of the myometrium (*p* = 0.004), and the longitudinal layers of the myometrium (*p* = 0.009), but did not differ significantly from the epithelial layer of the endometrium (*p* = 0.187). Likewise, sows with parity numbers ≥ 6 also exhibited a lower oxytocin receptor (H-score) than primiparous sows in most layers of the endometrium and myometrium of the uterus (*p* < 0.05), except in the epithelial layer of the endometrium (*p* = 0.400) and the longitudinal layers of the myometrium (*p* = 0.347).

Overall, the H-score of the oxytocin receptor in the uterine tissue did not differ between the sows in the luteal and follicular phases (*p* = 0.505). Similarly, the H-score of the oxytocin receptor in the five layers of uterine tissue did not differ between the luteal and follicular phases (*p* > 0.05) (Table 5).

## 4. Discussion

The present study was conducted to undertake an investigation into various aspects related to hyperprolific sows, with a particular focus on the morphology of their reproductive organs and the distribution of oxytocin receptors in uterine tissue across different parities and stages of the estrous cycle. The presence of oxytocin receptors in uterine tissue has been well established across several mammalian species [17,19,23]. In the present study, oxytocin receptor immunolocalization was detected in various cell types, including surface epithelial cells of the endometrium, endometrial glandular cells, myocytes of the myometrium, vascular endothelial cells, vascular smooth muscles, and endometrial stromal cells. Similar results have been reported in non-pregnant mares [19] and pregnant canines [23]. Among the five layers of the uterus, the present study identified the greatest H-score of oxytocin receptors in the deep uterine glands, while the circular muscle layer of the myometrium exhibited the lowest H-score. Within the endometrium, deep uterine glands had a greater H-score compared to superficial uterine glands. Additionally, there was no significant difference in H-score between the circular and longitudinal layers of the myometrium. A previous study in cyclic gilts demonstrated that the concentration of oxytocin receptor protein in the endometrium and myometrium did not differ significantly [16]. However, in both tissues, concentrations of the oxytocin receptor were reported to be lower during the luteal phase and increased during the follicular phase [16]. This finding differs from the current study, which found no significant difference in H-score between the luteal and follicular phases. While several studies have revealed factors influencing oxytocin receptor expression, such as the cycle phase, pregnancy, and specific hormones [16,24,25], none of them have studied the effects of aging on the oxytocin receptors in the porcine uterus. To our knowledge, the present study represents the first report on the effect of age, specifically in terms of parity, on the level of oxytocin receptors in the uterine tissue of hyperprolific sows. The current findings demonstrate that older sows (parity numbers ≥ 6) exhibited decreased oxytocin receptor abundance in most uterine layers compared to younger sows. This is a possible explanation for a previous study reporting a greater intensity of uterine contraction during parturition in sows with parity 1 compared to later parities, thus affecting farrowing duration in older sows [2,26]. A previous study suggested that the oxytocin receptor gene exhibited a moderate, constitutive, and cell-specific basal level of expression [27]. This expression can be further upregulated in specific situations, such as during full-term parturition. Conversely, it can also be specifically suppressed, for example, by circulating progesterone during certain phases of the reproductive cycle [27]. Furthermore, the current study suggests that aging may decrease the capability of specific cells to express the oxytocin receptor. However, further studies on the effect of aging on the function of the oxytocin receptor are needed to gain more detailed insight into this issue.

In commercial herds, the reproductive performance of sows is the main focus. It is normal practice to cull sows after their reproductive performance decreases, particularly in aged sows or those with a parity of more than 6 [5]. However, sow productivity today is boosted due to the use of high-prolificacy sow genetics [1]. A previous study reported significant correlations of ovarian, uterine, and vaginal characteristics with the mean lifetime numbers of live-born and stillborn piglets per litter and the last litter size before culling [7]. Additionally, ovarian and uterine sizes were used in a prior study to indirectly estimate uterine capacity in developing gilts [28]. The present study found that the average length of the uterus in Landrace × Yorkshire sows with parity 1–9 was similar to that of Large White sows from five decades ago [29]. Similarly, no significant difference in uterine length was observed between Landrace × Yorkshire sows with parity 1 and those reported three decades ago for Large White gilts [30]. These data suggest that uterine size has remained stable over time, while its capacity to accommodate larger litters has improved. Furthermore, the present study revealed positive correlations between sow age and various reproductive organ morphologies, including ovary weight, oviduct length, uterus weight, and the number of ovulations. We also observed a positive correlation between uterus length and sow body weight, which aligns with the growth curve reported by Solanes and Stern [31] for Large White female pigs. This curve reported that the live weight of Large White female pigs increased rapidly from 150 to 550 days of age and then slowed down after 750 days of age. This suggests that sows continue to grow rapidly until reaching parity 1, and, thereafter, their weight becomes relatively stable by parity 2–3. In terms of parity, the findings of the present study showed no significant difference in reproductive organ morphology among sows with parity numbers 2–5 and ≥6, except for oviduct length. However, compared to higher-parity sows, those with parity 1 exhibited lighter ovaries, a lighter and shorter uterus, and fewer ovulations. These findings align with previous studies indicating a tendency for changes in uterine horn length with age [29] and stabilized dimensions of the reproductive tract, particularly the vagina, after the first parity [32]. The present study found that the weight of the uterus in the follicular phase group was heavier than that in the luteal phase group, presumably reflecting estrogenic activity. Despite the heavier weight of the uterus in the follicular phase group, its length tended to be shorter than that in the luteal phase group. This finding aligns with a previous study [29] that observed that the uterine horns of sows were about twice as long in diestrus as they were in estrus. Additionally, Kaeoket et al. [33] reported variations in uterine weight among stages of the estrous cycle, with greater uterine weight found during late diestrus, proestrus, and estrus. This increased uterine weight was possibly due to uterine edema and a large number of capillaries underneath the surface epithelium during these stages of the estrous cycle [33].

A limitation of the current study was the use of samples collected from a slaughterhouse, which made it difficult to precisely determine the specific stage of the estrus cycle and the hormonal profile of the sows. Additionally, the number of samples in the follicular phase was limited due to its shorter duration compared to the luteal phase. Consequently, due to the limited number of follicular phase samples, a two-way analysis of parity and cycle phase group could not be performed.

## 5. Conclusions

In conclusion, the current study revealed significant differences in several aspects among sows with different parity numbers, including ovarian and uterine weight, ovulation rates, and uterine length. Notably, there was a distinct pattern of oxytocin receptor, as indicated by H-scores, particularly between sows with parity numbers 1 and ≥6. These findings highlight the complex interplay between parity, reproductive outcomes, and the oxytocin receptor in hyperprolific sows.

## Figures and Tables

**Figure 1 animals-14-01846-f001:**
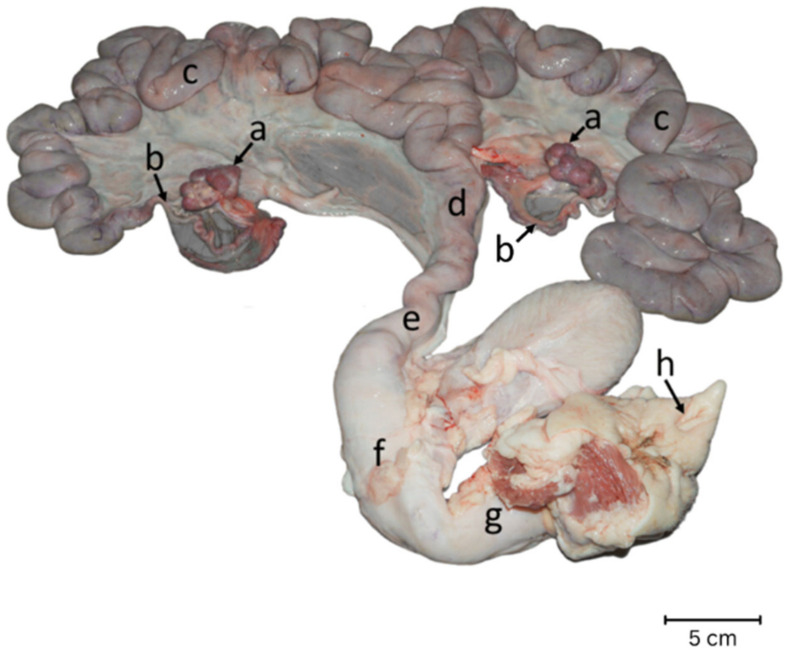
Gross morphology of the reproductive organs of Landrace × Yorkshire sows. a = ovary, b = oviduct, c = uterine horn, d = uterine body, e = cervix, f = vagina, g = vestibule, and h = vulva.

**Figure 2 animals-14-01846-f002:**
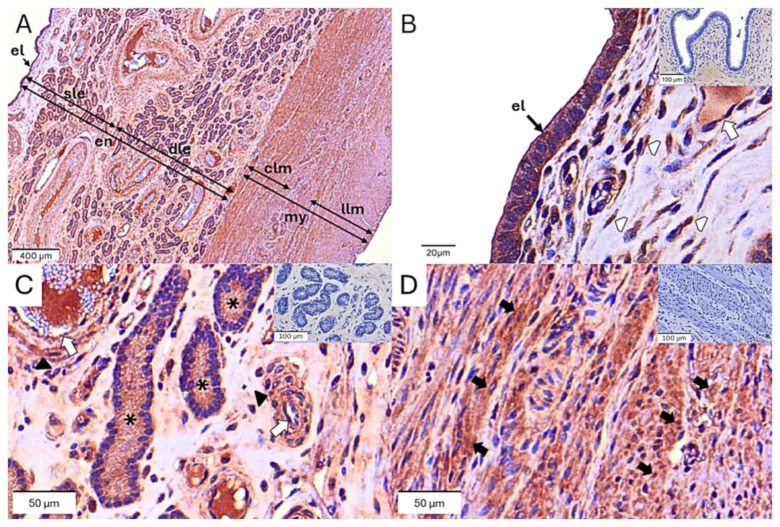
Immunohistochemical localization of the oxytocin receptor in the sow uterus. (**A**) Oxytocin receptor immunolocalization is observed in several layers of sow uterine tissue, including the surface epithelial layer (el), superficial (sle), and deep layers (dle) of the endometrium (en), as well as the circular (clm) and longitudinal layers (llm) of the myometrium (my). (**B**–**D**) Strong intensity of DAB staining is found in the surface epithelial cells (el), glandular epithelial cells (star), myocytes of the myometrium (black arrow), vascular endothelial cells (open arrow), vascular smooth muscles (head arrow), and endometrial stromal cells (open head arrow) of the uterine tissue. There is no background staining in the negative control (insert to **B**–**D**).

**Table 1 animals-14-01846-t001:** Descriptive statistics on sow reproductive performance at the last farrowing and gross morphology of reproductive organs.

Variables	Mean ± SD	Range
Sow data (*n* = 32)		
Age at culling (d)	799.8 ± 327.8	463.0–1550.0
Parity number at culling	3.4 ± 2.3	1.0–9.0
Body weight at culling (kg)	213.2 ± 31.7	139.5–281.5
Total number of piglets born per litter	15.5 ± 4.8	4.0–26.0
Stillbirths (%)	7.6 ± 9.3	0.0–29.4
Litter weight at birth (kg)	15.4 ± 6.3	0.9–25.1
Reproductive organs morphometry (*n* = 32)		
Number of ovulations	26.3 ± 7.3	14.0–40.0
Weight of ovary (g)	12.6 ± 6.0	4.7–31.9
Length of oviduct (cm)	33.5 ± 7.2	22.3–59.5
Length of uterus (cm)	348.3 ± 88.6	186.0–545.5
Weight of uterus (g)	1071.9 ± 446.6	258.0–2296.0
Length of cervix (cm)	24.5 ± 3.6	18.0–34.0
Length of vagina (cm)	13.0 ± 2.5	8.5–20.0
Length of vestibule (cm)	10.8 ± 1.9	7.0–17.0

**Table 2 animals-14-01846-t002:** Spearman’s correlations between the age and body weight of sows with their reproductive performances at the last farrowing and the gross morphology of their reproductive organs.

Variables	Correlation Coefficient (r)	
	Age of Sow (days)	Body Weight of Sow (kg)
Reproductive performance at the last farrowing (*n* = 32)		
Total number of piglets born per litter	NS	NS
Stillbirths (%)	NS	NS
Litter weight at birth (kg)	NS	NS
Reproductive organs morphometry (*n* = 32)		
Number of ovulations	0.432 *	NS
Weight of ovary (g)	0.559 ***	NS
Length of oviduct (cm)	0.413 *	NS
Length of uterus (cm)	NS	0.383 *
Weight of uterus (g)	0.487 **	NS
Length of cervix (cm)	NS	NS
Length of vagina (cm)	NS	NS
Length of vestibule (cm)	NS	NS

NS = not significant (*p* > 0.05); significance is indicated by * 0.01 < *p* < 0.05, ** 0.001 < *p* < 0.01, and *** *p* < 0.001.

**Table 3 animals-14-01846-t003:** Sow data and reproductive performance at the last farrowing among parity groups (*n* = 32) (least-square means ± SEM).

Variables	Parity			*p* Value
	1	2–5	≥6	
Number of sows	10	12	10	
Age at culling (d)	502.5 ± 38.8 ^a^	778.1 ± 38.5 ^b^	1272.7 ± 43.1 ^c^	<0.001
Parity number at culling	1.0 ± 0.0 ^a^	3.0 ± 0.3 ^b^	6.3 ± 0.3 ^c^	<0.001
Body weight at culling (kg)	205.7 ± 11.1	206.8 ± 10.7	216.5 ± 11.9	0.725
Total number of piglets born per litter	16.1 ± 1.6	14.2 ± 1.6	17.6 ± 1.7	0.251
Stillbirths (%)	6.0 ± 3.1	5.9 ± 3.1	11.5 ± 3.5	0.321
Litter weight at birth (kg)	16.8 ± 2.0 ^ab^	13.6 ± 2.2 ^a^	19.3 ± 2.2 ^b^	0.115

^a,b,c^ Values with different superscripts within rows differ significantly (*p* < 0.05).

**Table 4 animals-14-01846-t004:** Gross morphology of reproductive organs of sows by parity and cycle phase groups (least-square means ± SEM).

Variables	Parity			*p* Value	Cycle Phase		*p* Value
	1	2–5	≥6		Luteal	Follicular	
Number of sows	10	12	10		26	6	
Number of ovulations	19.9 ± 2.4 ^a^	29.7 ± 2.0 ^b^	27.7 ± 2.1 ^b^	0.013	-	-	
Weight of ovary (g)	7.1 ± 1.7 ^a^	12.5 ± 1.7 ^b^	14.6 ± 1.8 ^b^	0.008	13.1 ± 1.0	9.7 ± 2.1	0.159
Length of oviduct (cm)	32.9 ± 2.2 ^a^	33.4 ± 2.1 ^a^	40.0 ± 2.4 ^b^	0.038	32.6 ± 1.3	38.3 ± 2.7	0.069
Length of uterus (cm)	277.9 ± 26.1 ^a^	354.6 ± 25.9 ^b^	346.6 ± 29.0 ^ab^	0.075	358.6 ± 15.7	294.1 ± 32.9	0.090
Weight of uterus (g)	902.9 ± 112.5 ^a^	1442.1 ± 111.8 ^b^	1394.3 ± 125.1 ^b^	0.002	947.7 ± 67.5	1545.2 ± 141.8	<0.001
Length of cervix (cm)	22.8 ± 1.2	25.2 ± 1.2	25.3 ± 1.3	0.223	24.4 ± 0.7	24.4 ± 1.5	0.988
Length of vagina (cm)	11.9 ± 0.8	13.9 ± 0.8	13.6 ± 0.9	0.160	12.8 ± 0.5	13.4 ± 1.0	0.631
Length of vestibule (cm)	11.3 ± 0.6	11.7 ± 0.6	11.0 ± 0.7	0.653	10.5 ± 0.4	12.2 ± 0.7	0.049

^a,b^ Values with different superscripts within rows differ significantly (*p* < 0.05).

**Table 5 animals-14-01846-t005:** Immunohistochemistry of oxytocin receptor (H-score) in different tissue layers of the uterus in sows by parity and cycle phase groups (least-square means ± SEM).

Variables	Parity			*p* Value	Cycle Phase		*p* Value
	1	2–5	≥6		Luteal	Follicular	
Number of sows	9	11	7		24	3	
Overall H-score	249.0 ± 7.3 ^a^	262.4 ± 7.7 ^a^	210.9 ± 9.6 ^b^	<0.001	236.6 ± 4.0	244.9 ± 11.7	0.505
Endometrium							
Surface epithelial cells	257.8 ± 18.8	266.7 ± 19.9	238.7 ± 24.7	0.524	255.1 ± 8.9	253.6 ± 26.1	0.957
Superficial uterine glands	244.7 ± 17.3 ^a^	241.4 ± 18.3 ^a^	188.4 ± 22.7 ^b^	0.048	208.4 ± 8.9	241.2 ± 26.1	0.236
Deep uterine glands	280.1 ± 11.7 ^a^	293.7 ± 12.3 ^a^	235.9 ± 15.3 ^b^	0.003	258.5 ± 8.9	281.3 ± 26.1	0.411
Myometrium							
Circular layer	231.5 ± 16.6 ^a^	244.2 ± 17.6 ^a^	181.6 ± 21.8 ^b^	0.024	225.5 ± 8.9	212.7 ± 26.1	0.642
Longitudinal layer	231.1 ± 16.3 ^ab^	265.8 ± 17.2 ^a^	209.7 ± 21.4 ^b^	0.036	235.5 ± 8.9	235.5 ± 26.1	0.999

^a,b^ Values with different superscripts within rows differ significantly (*p* < 0.05).

## Data Availability

The data presented in this study are available on request from the corresponding author.

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
