# Peer review of "Exploring Hyperprolific Sows: A Study of Gross Morphology of Reproductive Organs and Oxytocin Receptor Distribution across Parities"

_animals, 2024, doi:10.3390/ani14131846_

Round 1

Reviewer 1 Report

Comments and Suggestions for Authors

The purpose of the study by Adi et al. is to evaluate the morphological organs of the female reproductive system of pigs defined as hyperprolific, that is, those that give birth to more piglets than they have functional nipples. This topic is moderately interesting from a scientific point of view. An additional contribution to science is the immunohistochemical evaluation of the presence of the oxytocin receptor in organs of the porcine reproductive system.

Specific comments:

l. 124 and the rest of the text - the uterus is a single organ

L. 125 - describe the method of animal death

L. 128 - the term expression refers only to genes and should only be used in that context

L. 129 - the uterus is an organ and not a tissue

L. 163 - how were organ samples fixed?

L. 176 - a negative control is not enough to check the specificity of the primary antibody used. Preadsorption tests are necessary.

Figure 2 - larger magnification of the images will definitely make it easier for the reader to understand what they are observing.

Author Response

Reviewer #1:

The purpose of the study by Adi et al. is to evaluate the morphological organs of the female reproductive system of pigs defined as hyperprolific, that is, those that give birth to more piglets than they have functional nipples. This topic is moderately interesting from a scientific point of view. An additional contribution to science is the immunohistochemical evaluation of the presence of the oxytocin receptor in organs of the porcine reproductive system.

Specific comments:

  1. 124 and the rest of the text - the uterus is a single organ

Answer: The term 'uteri' has been changed to 'uterus' throughout the manuscript. Thank you.

  1. 125 - describe the method of animal death

Answer: Additional method concerning animal death has been added: “The sows were slaughtered following standard procedures at a commercial slaughterhouse. This involved restraining them in a manner designed to minimize stress, using electrical stunning to render them unconscious and insensible to pain. Following stunning, the sows' throats were cut to facilitate bleeding out, after which the carcasses underwent further processing for meat production, including skinning and removal of internal organs.”

  1. 128 - the term expression refers only to genes and should only be used in that context

Answer: We have revised accordingly and used the term “immunolocalization” instead of “expression” in our results.

  1. 129 - the uterus is an organ and not a tissue

Answer: ‘the uterine tissue’ has been changed to ‘the uterus’.

  1. 163 - how were organ samples fixed?

Answer: Additional information concerning fixation has been added: “The tissue samples of the uterus were fixed in 4% paraformaldehyde for 24–48 h, processed with an automatic tissue processor (Tissue-Tek VIP 5 Jr., Sakura, Tokyo, Japan), and embedded in paraffin blocks (Tissue-Tek TEC, Sakura, Tokyo, Japan). The paraffin blocks were then cut into 5-µm-thick sections using a microtome (Shandon, Anglia Scientific Instrument Ltd., Cambridge, UK).”

  1. 176 - a negative control is not enough to check the specificity of the primary antibody used. Pre-adsorption tests are necessary.

Answer: This is a very useful technical comment. However, in this experiment, a pre-adsorption test was not conducted. However, we followed the manufacturer's protocol for immunohistochemical detection of the oxytocin receptor and employed the same technique for evaluating the negative control as used in previously published studies (Prapaiwan et al. 2017, Theriogenology 100: 59-65; Phoophitphong et al., 2017, Anat. Histol. Embryol. 46: 334–341).

Figure 2 - larger magnification of the images will definitely make it easier for the reader to understand what they are observing.

 Answer: We have changed the figure to a higher magnification.

Reviewer 2 Report

Comments and Suggestions for Authors

In the present study, the authors have evaluated the gross morphology of reproductive organs and the distribution of oxytocin receptors across different parities in hyperprolific sows. The study is well-defined and structured, providing new insights into the handling of sows during parturition across different parities. However, there are minor observations that need incorporation into the respective sections. The article has the potential for publication in the journal Animal.

Following are some queries after reviewing the submission:

Title:

  • It is okay and catchy.

Simple summary:

  • Well-described.

Abstract:

  • It is well-presented but can be reduced by deleting some unnecessary lines.

Introduction:

  • This part has been presented with enough information about problems linked to proliferative sow lines. At the start, a few lines should be included about the advantages of proliferative sows in the swine industry. The second paragraph should be written to explain what is already known and what needs to be done.

Materials and Methods:

  • The contents are clearly demonstrated.
  • How were tissues collected and processed before IHC? This needs to be added.

Results:

  • Only correlations of age and body weight with other parameters were presented. What about the correlations of other studied variables?
  • Authors should include a correlations table instead of general statistics of parameters.
  • The number of sows for luteal and follicular phases in Table 3 is not given.
  • In my opinion, the organ morphometry with respect to parity and cyclicity should be presented in a single table.
  • The same comment applies to the oxytocin receptor values.
  • Table 4 is not showing any oxytocin receptor variation in different tissues but is described in the text (lines 248-252).
  • If the authors have tissue leftovers, then quantification of oxytocin receptors can validate the results related to parity.

Discussion:

  • This part is up to the mark, justifying the results and discussing them in a scientific manner.
  • What were the limitations of this study?

Conclusion:

  • Reflects the main outcomes of the study.

Author Response

In the present study, the authors have evaluated the gross morphology of reproductive organs and the distribution of oxytocin receptors across different parities in hyperprolific sows. The study is well-defined and structured, providing new insights into the handling of sows during parturition across different parities. However, there are minor observations that need incorporation into the respective sections. The article has the potential for publication in the journal Animals.

Following are some queries after reviewing the submission:

Title:

It is okay and catchy.

Simple summary:

Well-described.

Abstract:

It is well-presented but can be reduced by deleting some unnecessary lines.

Answer: Thank you very much for your kind suggestion. The manuscript has been revised according to both the reviewer’s and the editor’s comments. All changes in the manuscript are indicated using 'blue' text. The abstract has been modified in response to Reviewer 1's comments; however, we have not deleted any lines and have retained all the important information.

Introduction:

This part has been presented with enough information about problems linked to proliferative sow lines. At the start, a few lines should be included about the advantages of proliferative sows in the swine industry. The second paragraph should be written to explain what is already known and what needs to be done.

Answer: We have added the advantages of prolific sows in the swine industry at the beginning of the introduction. Information in the second paragraph has already mention what is already known and what needs to be done.

Materials and Methods:

The contents are clearly demonstrated.

How were tissues collected and processed before IHC? This needs to be added.

Answer: The collection and processing of tissue samples before IHC have been added to the 'Materials and Methods' section under subheading 2.3, Immunohistochemical Staining.

Results:

Only correlations of age and body weight with other parameters were presented. What about the correlations of other studied variables?

Authors should include a correlations table instead of general statistics of parameters.

Answer: Additional correlations table has been added to the revised manuscript.

The number of sows for luteal and follicular phases in Table 3 is not given.

Answer: We have added the number of sows in the Table 3.

In my opinion, the organ morphometry with respect to parity and cyclicity should be presented in a single table.

Answer: We have combined organ morphometry with respect to parity and cyclicity into one Table (Table 4).

The same comment applies to the oxytocin receptor values.

Answer: Modified as suggested (Table 5).

Table 4 is not showing any oxytocin receptor variation in different tissues but is described in the text (lines 248-252).

Answer: We have omitted the table citation in this part. Thank you for your correction.

If the authors have tissue leftovers, then quantification of oxytocin receptors can validate the results related to parity.

Answer: We still have tissue samples preserved in paraformaldehyde solution. However, the tissues were not cryopreserved in this study. It could also be interesting to quantify oxytocin receptors using a different method. In the future, we plan to analyze other hormone receptors related to reproduction. Thank you for your suggestions.

Discussion:

This part is up to the mark, justifying the results and discussing them in a scientific manner.

What were the limitations of this study?

Answer: Additional limitation of the study has been addressed in the discussion: “A limitation of the current study was the use of samples collected from a slaughterhouse, which made it difficult to precisely determine the specific stage of the estrus cycle and the hormonal profile of the sows. Additionally, the number of samples in the follicular phase was limited due to its shorter duration compared to the luteal phase. Consequently, due to the limited number of follicular phase samples, a two-way analysis of parity and cycle phase group could not be performed.”

Conclusion:

Reflects the main outcomes of the study.

Answer: Thank you very much for all your comments. We have revised the manuscript accordingly.